# Clinical Features and Outcomes of Primary Cutaneous Peripheral T-Cell Lymphoma, Not Otherwise Specified, Treated with CHOP-Based Regimens

**DOI:** 10.3390/cancers17101673

**Published:** 2025-05-15

**Authors:** Ge Hu, Zheng Song, Chao Lv, Yifei Sun, Yidan Zhang, Xia Liu, Xue Han, Lanfang Li, Lihua Qiu, Zhengzi Qian, Shiyong Zhou, Wenchen Gong, Bin Meng, Jin He, Xianhuo Wang, Huilai Zhang

**Affiliations:** 1State Key Laboratory of Druggability Evaluation and Systematic Translational Medicine/Department of Lymphoma, Tianjin Medical University Cancer Institute and Hospital, National Clinical Research Center for Cancer, Tianjin’s Clinical Research Center for Cancer, Key Laboratory of Cancer Prevention and Therapy, The Sino-US Center for Lymphoma and Leukemia Research, Tianjin 300060, China; hugu1009@163.com (G.H.); songzheng_tjzlyy@163.com (Z.S.); 17275451992@163.com (C.L.); sunyf451302@163.com (Y.S.); aria0411@163.com (Y.Z.); tjxialiu@163.com (X.L.); hanxue0453@163.com (X.H.); lilanfangmeng@163.com (L.L.); tjlihuaqiu@163.com (L.Q.); tjzhengziqian@163.com (Z.Q.); zsy1003@163.com (S.Z.); tjzlyy_xianhuow@163.com (X.W.); 2Department of Pathology, Tianjin Medical University Cancer Institute and Hospital, Tianjin 300060, China; tjwchgong@163.com (W.G.); mengb321@163.com (B.M.)

**Keywords:** cutaneous, peripheral T-cell, lymphoma, extranodal, PTCL-NOS

## Abstract

Primary cutaneous peripheral T-cell lymphoma, not otherwise specified (pcPTCL-NOS), is a rare, aggressive, fatal type of cutaneous T-cell lymphoma that accounts for only approximately 2% of primary cutaneous lymphomas. This study examined the clinical manifestations, immunophenotypic characteristics, selection of treatment, and outcomes of patients with pcPTCL-NOS. Fifteen patients with pcPTCL-NOS were here identified. A detailed selection of treatments and combination applications of new drugs for these patients were described. All fifteen patients were treated with CHOP-based regimens as the initial treatment. Generally, pcPTCL-NOS requires early and active systemic treatment. However, for patients with T1 tumors, reducing the intensity of treatment with CHOP should be appropriately considered.

## 1. Introduction

Cutaneous T-cell lymphomas (CTCLs) are a heterogeneous group of cutaneous lymphoproliferative disorders (LPDs) characterized by the neoplastic proliferation of clonal T-cells in the skin [1,2]. CTCL represents approximately 75–80% of all primary cutaneous lymphomas. The World Health Organization–European Organization for Research and Treatment of Cancer (WHO–EORTC) provides a classification of cutaneous lymphomas based on the clinical, pathological, and molecular characteristics, and mycosis fungoides (MF) and CD30+ LPD are the most frequent entities. All other cases, which cannot be assigned to other specific lymphoma categories, are referred to as primary cutaneous peripheral T-cell lymphoma, not otherwise specified (pcPTCL-NOS) or unspecified. With the increasing number of non-mycosis fungoides cutaneous T-cell lymphomas (non-MF CTCLs) observed in recent years, the frequency of pcPTCL-NOS has decreased. pcPTCL-NOS is a rare, aggressive, fatal type of cutaneous T-cell lymphoma (CTCL) that accounts for only approximately 2% of primary cutaneous lymphomas [3,4]. The clinical presentation of pcPTCL-NOS is characterized by solitary, localized or, more frequently, generalized plaques, nodules, or tumors. The 5-year survival rate for patients with pcPTCL-NOS is less than 20% [5,6], which makes both a timely diagnosis and effective treatment important measures to improve outcomes.

Owing to the rarity of pcPTCL-NOS, a paucity of data regarding its clinicopathological and immunophenotypic features, clinical course, and outcomes exists. In accordance with the World Health Organization–European Organization for Research and Treatment of Cancer (WHO–EORTC, 2018) and the World Health Organization classification (5th edition, 2022) [4,7], we conducted a real-world study to examine the clinical manifestations, immunophenotypic characteristics, selection of treatment, and outcomes of patients with pcPTCL-NOS.

## 2. Methods

### 2.1. Patients and Data Collection

Patients who were diagnosed with pcPTCL-NOS according to the World Health Organization–European Organization for Research and Treatment of Cancer (WHO–EORTC, 2018), as well as the World Health Organization classification (5th edition, 2022), between January 2014 and August 2024 at Tianjin Medical University Cancer Institute and Hospital (TMUCIH), were included in this study. Patients with extracutaneous involvement at the time of diagnosis were excluded; by definition, they represented secondary cutaneous involvement of nodal PTCL-NOS. Two pathologists reviewed the disease pathology for all patients. In this retrospective chart review, data on the baseline demographic characteristics, laboratory parameters at diagnosis, treatment regimens, and patient follow-up were collected.

Staging was performed according to the instructions of the International Society for Cutaneous Lymphomas/European Organization of Research and Treatment of Cancer (ISCL/EORTC) Proposal on the TNM Classification of Cutaneous Lymphomas other than MF/SS [8]. The responses to the therapy were evaluated according to the 2014 Lugano criteria. The overall response rate (ORR) was defined as the proportion of patients who achieved a complete response (CR) or partial response (PR), and the complete response rate (CRR) was the proportion of patients who achieved a CR. Disease relapse or progression was defined as the detection of a new lesion or the progression of an existing lesion according to a physical or radiographic examination. Progression-free survival (PFS) was defined as the time from the initial treatment to disease recurrence, disease progression, the last follow-up, or death. OS was defined as the time from the initial treatment until death from any cause or the last follow-up date. Patients were followed by referring to medical records and by telephone, and the last follow-up date was August 2024.

Approval for this study was granted by the institutional review board at TMUCIH, and this study was conducted in accordance with the Declaration of Helsinki. Informed consent was obtained from all patients.

### 2.2. Statistical Analysis

Statistical analyses were performed with SPSS 26.0 and R4.1.0 software. Continuous variables are presented as medians, interquartile ranges (IQRs), and ranges; categorical variables are presented as frequency counts and percentages. Owing to the generally skewed distributions of all the variables, nonparametric tests were performed. PFS and OS were evaluated by means of the Kaplan–Meier method with the log-rank test. Cox proportional hazards regression models were used to estimate hazard ratios (HRs) and 95% confidence intervals (CIs). Statistical significance was set at *p* < 0.05.

## 3. Results

Fifteen patients with pathologically confirmed pcPTCL-NOS between January 2014 and August 2024 at TMUCIH were identified. The total clinicopathological characteristics of the 15 patients with pcPTCL-NOS are summarized in Appendix A.

### 3.1. Clinical and Pathological Features

The median age at diagnosis was 54 years (range: 31–72 years), and 66.67% of patients were under 60 years old. A preponderance of females was observed, with 11/15 (73.33%) being women and 4/15 (26.67%) being men. At the time of diagnosis, 13.33% (*n* = 2) of the patients had B symptoms, and elevated beta 2-microglobulin and lactate dehydrogenase (LDH) levels were found in 26.67% (*n* = 4) of the patients. Five out of fifteen patients (33.33%) presented with a solitary nodule/tumor according to the TNM stage (stage T1), and four out of fifteen patients (26.67%) presented with localized papules and nodules (stage T2), whereas six out of fifteen patients (40%) presented with disseminated papules and nodular lesions (stage T3). The lesion sites in the patients who presented with a solitary nodule/tumor (T1) were the head (three out of five patients), arm (one out of five patients), and leg (one out of five patients), whereas those in the patients with localized papules and nodules (T2) were the arms (two out of four patients), head (one out of four patients), and hand (one out of four patients). In patients with disseminated disease, the lesions were mostly located on the trunk and extremities. The clinical features of each patient are summarized in Table 1.

Among the 15 patients, the details of the CD4/CD8 immunophenotypic characteristics were as follows: CD4+/CD8− in 53.33% (8/15 patients), CD4+/CD8+ in 26.67% (4/15 patients), CD4−/CD8− in 13.33% (2/15 patients), and CD4−/CD8+ in 6.67% (1/15 patients). The phenotypes varied across stages as follows: stage T1, CD4+/CD8− in 20% (1/5 cases), CD4+/CD8+ in 40% (2/5 cases), CD4−/CD8− in 20% (1/5 cases), and CD4−/CD8+ in 20% (1/5 cases); stage T2, CD4+/CD8− in 75% (3/4 cases) and CD4+/CD8+ in 25% (1/4 cases); and stage T3, CD4+/CD8− in 66.67% (4/6 cases), CD4+/CD8+ in 16.67% (1/6 cases), and CD4−/CD8− in 16.67% (1/6 cases). In this group, only one patient with a stage T1 tumor had a CD4−/CD8+ phenotype (Figure 1). Ki67 staining revealed proliferative fractions of ≥80% and <80% in 40% (6/15 patients) and 60% (9/15 patients), respectively. The percentages of CD30- and CD20-positive patients were 40% (6/15 patients) and 33.33% (5/15 patients), respectively. The percentage of PD-1-positive patients was 20% (3/15 patients). One patient with a stage T3 tumor had a T follicular helper (TFH) cell phenotype of CD10+, CXCL13+, PD1+, and BCL6+. The presence of EBV, as assessed by in situ hybridization for EBER, was negative in all patients, with one patient with a stage T3 tumor testing positive for EBV-DNA. The histological and immunophenotypic characteristics of each patient are summarized in Table 2.

### 3.2. Frontline Treatments and Responses

All fifteen patients with pcPTCL-NOS received systemic chemotherapy as the initial treatment. The effects of combination therapy as a first-line treatment are shown in Figure 2. The frontline treatment was the combination of cyclophosphamide, doxorubicin, vincristine, and prednisone (CHOP) or CHOP-like regimens. Three patients with stage T1 tumors underwent complete lesion resection before chemotherapy, seven patients (four with both stage T1 and T2 tumors and three with stage T3 tumors) received CHOP combined with chidamide (tucidinostat), two patients with stage T3 tumors received brentuximab vedotin, two patients received CHOP combined with mitoxantrone liposomes (Lipo-Mit) (one patient with a stage T1 tumor and the other with a stage T3 tumor), three patients received CHOP combined with radiotherapy (two patients with stage T1 tumors and one patient with a stage T3 tumor), and two patients (both with stage T3 tumors) received ASCT after the first-line treatment. The initial response to treatment was complete remission (CR) in nine patients (60%) and partial remission (PR) in four patients (26.67%), and the ORR was 86.67%.

Among the nine patients who achieved a CR after the first-line treatment, five had stage T1 tumors, four had stage T3 tumors, and none had stage T2 tumors. Notably, two patients received the CMOPE regimen, in which mitoxantrone liposomes (Lipo-Mit) replaced anthracyclines in the CHOPE regimen, one of whom had a stage T1 tumor and the other had a stage T3 tumor; both achieved a CR after the first-line treatment, and, currently, they are still in CR. Four patients (two each with stage T1 and T3 tumors) received CHOP or CHOP-like regimens combined with chidamide, and one (with a stage T3 tumor) received chidamide combined with brentuximab vedotin.

Among the four patients who achieved a PR after first-line treatment, three had stage T2 disease, one had stage T3 disease, and none had stage T1 disease. Two patients with stage T2 disease received CHOP or a CHOP-like regimen alone, one patient with stage T2 disease received CHOP combined with chidamide, and another patient with stage T3 disease received CHOP combined with chidamide and BV simultaneously.

Two patients experienced disease progression after the first-line treatment, with one patient with stage T2 disease progressing after receiving six cycles of chidamide and the CHOPE regimen and the other patient with stage T3 disease progressing after receiving only two cycles of the CHOPE regimen. The detailed first-line treatments and responses of each patient are listed in Table 3.

### 3.3. Salvage Treatments and Responses

In total, eight patients relapsed and received salvage treatment, including 20% of the patients (1/5) with stage T1 tumors, 100% of the patients (4/4) with stage T2 tumors, and 50% of the patients (3/6) with stage T3 tumors. Various attempts were made to treat patients who experienced progression or relapse. In addition to second-line chemotherapy, such as the GDP, GEMOX, DICE, and DHAP regimens, patients chose immunochemotherapy, targeted therapy, or participation in clinical trials (involving agents such as chidamide, brentuximab vedotin, mitoxantrone liposomes, PD1 inhibitors, JAK1 inhibitors, and azacitidine). However, salvage treatment failed in three patients (two patients with stage T2 tumors and one with a stage T3 tumor), and the patients died due to disease progression. Ultimately, the response to salvage treatment was a CR in three patients (3/8, 37.5%) and PR in one patient (1/8, 12.5%), and the ORR was 50% (4/8).

Two patients received treatment with the combination of chidamide and azacitidine: one patient with a stage T1 tumor achieved a CR again, and the other patient with a stage T3 tumor died due to disease progression. One patient with a stage T2 tumor previously received a six-cycle CHOPE regimen as the first-line chemotherapy and then a one-cycle GDP regimen as a second-line treatment, but the disease had not been effectively controlled; salvage therapy included DICE combined with the chidamide regimen and the patient ultimately achieved a CR. One patient with a stage T2 tumor experienced central nervous system involvement after three cycles of treatment with CHOPE combined with the chidamide regimen as the first-line treatment; this patient tested positive by cerebrospinal fluid flow cytometry. Despite receiving high-dose MTX treatment and the DHAP regimen as the salvage treatment, this patient died due to disease progression. One patient with a stage T3 tumor expressing PD-1 who received tislelizumab (PD-1 inhibitor) as the second-line therapy achieved a CR again. Another patient with a stage T3 tumor received golidocitinib (JAK1 inhibitor) combined with the chidamide regimen as a second-line therapy for two cycles at the last follow-up date, and the disease remained stable. Two patients with stage T2 disease received CHOP or CHOP-like regimens alone as the first-line treatment; one of them died after four cycles of treatment with the GDP regimen as the second-line therapy, and the other patient achieved disease control after treatment with BV combined with chidamide. The detailed salvage treatments, responses, and survival data for each patient are listed in Table 3.

### 3.4. Survival and Prognostic Factors

With a median follow-up of 40 (range: 5–105) months, the median PFS was 21 months, and the median OS was not reached. The OS rates at 1 year, 2 years, and 3 years were 80%, 77.8%, and 77.8%, respectively, and the PFS rates were 60%, 44.4%, and 33.3%, respectively (Figure 3).

An analysis of the survival data revealed that 20% (3/15) of all patients died due to lymphoma (DDL). Twenty percent (3/15) of the patients were alive with disease (AWD), and 60% (9/15) were alive without disease (AWOD) at the last follow-up. The outcomes according to the tumor stage were as follows: for T1 (solitary lesion), all five patients were alive without disease (AWOD), and for T2 (localized disease), one patient (25%, 1/4) was alive without disease (AWOD), one patient (25%, 1/4) was alive with disease (AWD), and two patients (50%, 2/4) died due to lymphoma (DDL). For T3, three patients (50%) were alive without disease (AWOD), one patient (33.33%, 2/6) was alive with disease (AWD), and one patient (16.67%, 1/6) died due to lymphoma (DDL). These group differences did not reach statistical significance.

The results of the univariate analyses indicated that patients with B symptoms and the CD4−/CD8− phenotype had inferior outcomes (*p* < 0.05). Age, sex, tumor stage, PIT score (calculated from 0 to 4 by age, performance status, lactic dehydrogenase level, and bone marrow involvement), Ki-67 index, elevated β2-MG levels, expression of CD20 or PD1, and selection of treatment were not associated with the prognosis (Table 4). A difference in survival trends was observed between the stage T1, T2, and T3 groups, but the difference was not significant because of the small sample size (Figure 4). We further analyzed survival according to different body regions and found a difference in survival trends between the solitary (T1) and disseminated (T2;T3) groups (Figure 5). These findings suggest that the intensity of treatment can be reduced for patients’ T1 tumors in the future.

## 4. Discussion

To date, this is the largest series of patients with pcPTCL-NOS according to the WHO–EORTC (2018), as well as the WHO (5th edition, 2022) classification, in a single tertiary center. In our series, in addition to clinical manifestations, immunophenotypic characteristics, outcomes, and prognoses, we provided a detailed description of the selection of treatments and combination applications of new drugs for patients with pcPTCL-NOS.

Cutaneous T-cell lymphoma (CTCL) has an indolent clinical course, and 20% to 55% of CTCL patients develop late-stage disease [9,10]. Most early-stage patients who do not require systemic chemotherapy seek treatment in the Department of Dermatology. In our center, all 15 patients with pcPTCL-NOS received systemic chemotherapy as the initial treatment since the disease course was expected to be aggressive. In contrast to previous reports [11,12,13], patients in this group were predominantly female (2.75:1), and the median age at diagnosis was 54 years, with 66.67% of patients being under 60 years old. The most common clinical presentations were nodules/tumors and papules and, less often, ulcers, which was consistent with the results of previous studies [14,15,16,17,18]. Notably, lesions on the head, mainly the facial areas, were more common in this group of patients than in previous studies [11]. These lesions can also lead to early detection and attention to the disease. Patients with B symptoms had inferior outcomes. In our series, a difference in survival trends was observed among the stage T1, T2, and T3 groups. The small number of patients may, however, be one reason for the lack of a significant difference in the survival curves. In addition, in our series, the predominant phenotype was CD4+/CD8−, followed by a double-positive phenotype and, less commonly, a double-negative phenotype. Patients with the CD4−/CD8− phenotype had inferior outcomes. In Kempf’s cohort [11], aberrant expression of CD20 was observed in approximately one-fifth of the tumors. We also detected aberrant CD20 expression in one-third of the patients (5/15). The difference was that two patients who received chemotherapy plus rituximab were included in their cohort, but no such patients were included in our study. In previous studies [19], CD20 itself was reported to be a dynamic marker that may either be gained or lost during the clonal evolution of a tumor. However, whether rituximab plus chemotherapy provides a clinical benefit in this situation is not yet clear [20]. CD30 expression has been consistently reported in CD30+ lymphoproliferative disorders (LPDs) but differs in other subtypes of cutaneous T-cell lymphoma [21]. BV may be a therapeutic option, even in patients with low or even no expression of CD30, but the duration of the response can be longer in patients with high CD30 expression [22,23,24]. In our study, six patients were CD30-positive, three of whom were treated with BV (two patients with stage T3 tumors treated in the first-line setting, and another with a stage T2 tumor treated in the second-line setting), and the ORR was 100%. Additionally, among cutaneous T-cell lymphomas (CTCLs), the classification does not yet include a specific category for those with a TFH phenotype. In our group, one patient with a stage T3 tumor presenting a T follicular helper (TFH) phenotype received the CHP regimen combined with BV and chidamide as the first-line therapy for four cycles, and was in a state of disease control at the last follow-up. As targeted treatments, including histone deacetylase inhibitors [25,26] and hypomethylating agents [27,28], have been introduced for systemic TFH lymphomas, therapeutic options for these patients can be explored. In their cohort, Kempf W et al. [11] also reported a patient with the TFH phenotype of pcPTCL-NOS. A genuine TFH phenotype seems to be rare in patients with pcPTCL-NOS.

With respect to treatment modalities, all patients in our study received systemic chemotherapy. Previous studies have suggested that immediate, intense treatment with multiagent chemotherapy and hematopoietic stem cell transplantation is indicated for patients with pcPTCL-NOS [18]. In our series, all five patients with stage T1 disease also received CHOP or CHOP-like regimens for four–six cycles as the frontline treatment with complete resection of the lesion and/or radiotherapy. With a median progression-free survival (PFS) of 19 months, all of the patients with stage T1 tumors were alive without disease. Compared with previous studies [11,12,13], for patients with stage T1 or even stage T2 disease, excision, focal radiotherapy, or excision followed by focal radiotherapy were the most common upfront strategies. The question of whether an aggressive therapeutic approach with chemotherapy in patients with single lesions is effective and necessary or justified at all, which was raised by previous research [11], requires further confirmation. In 2022, Stuver R et al. [13] reported a group of patients with pcPTCL-NOS in which patients with a single lesion or regional disease experienced significantly longer survival than those with multiple regions involved. Therefore, systemic chemotherapy may not be necessary for patients with stage T1 disease, and surgery and/or radiotherapy alone are likely sufficient. Among the eight patients who relapsed after the first-line treatment, 50% (4/8) had stage T2 disease. Compared with patients with stage T2 disease, patients with stage T1 disease may also undergo surgical resection and local radiation therapy, and patients with stage T3 disease can more easily receive treatment with a new drug and even autologous stem cell transplantation. This finding may explain why recurrence is more common in patients with stage T2 disease. This finding also suggests that the treatment for patients with stage T2 disease may be insufficient.

A clinical study initiated by our center regarding azacitidine combined with chidamide for patients with R/R peripheral T-cell lymphoma is ongoing, and two patients received this regimen as salvage treatment. One patient with stage T1 disease achieved a CR, while the other patient with stage T3 disease experienced disease progression and died due to lymphoma. The preliminary results of this study indicate that the application of the chidamide and azacitidine regimen as a salvage treatment in patients with advanced tumors may not be sufficient. However, more clinical data are needed to confirm these findings. Two patients received the CMOPE regimen as an initial treatment, and achieved CR. This was a phase I study initiated by our center, and showed promising anti-tumor activity, evidenced by an ORR of 100% and a CR rate of 66.7%, as first-line therapy in patients with untreated PTCL [29].

Two patients with stage T2 disease received CHOP or CHOP-like regimens alone as a first-line treatment, and the efficacy evaluations revealed that both regimens resulted in a PR. The difference is that one patient was switched to the GDP regimen for subsequent treatment, whereas the other patient continued to receive the CHOP-like regimen in combination with two new drugs (chidamide and brentuximab vedotin). Finally, the former died due to disease progression, whereas the latter was alive with disease at the last follow-up. From this perspective, the application of new drugs, and even combinations of drugs with multiple mechanisms, may further relieve disease in patients.

Notably, in this group of patients, one patient with stage T2 disease had central nervous system involvement. The patient experienced headaches and increased skin lesions and died due to lymphoma. Involvement of the central nervous system in patients with CTLC is rare. Further consideration is needed to determine whether it occurs simultaneously or earlier with respect to skin lesions. A physical examination and imaging examination of the central nervous system should be performed in clinical practice. Simultaneously, two patients with stage T3 disease received ASCT after the first-line treatment. One patient remained in a CR, while the other patient experienced disease recurrence. The patient with disease progression received chidamide combined with the CHOPE regimen as the first-line treatment, and immunohistochemistry showed CD30-negative and PD1-positive expression. Thus, a PD-1 inhibitor (tislelizumab) was selected for this patient as the second-line treatment. After 18 cycles of treatment, the patient achieved a CR again. PD-1, a membrane receptor expressed on activated T-cells, inhibits the immune response in peripheral tissues and promotes self-tolerance. In CTCL, skin lesion-derived tumor-infiltrating T-cells have higher expression of PD-1 (and other immune checkpoint molecules) in comparison to controls [30]. Nevertheless, ongoing studies should define more clearly the efficacy and safety of PD1 inhibitors in the T-cell lymphoma treatment.

Additionally, in China, chidamide has been approved for the treatment of relapsed/refractory peripheral T-cell lymphoma (PTCL) for 10 years, and its efficacy is recognized in PTCL patients [31,32]. In our study, most patients were treated with a combination of chidamide (seven patients in the first-line group and three patients in the salvage treatment group). Two patients died, and the remaining eight patients achieved disease control. Another patient with stage T3 disease received golidocitinib (JAK1 inhibitor) combined with the chidamide regimen as the second-line therapy for two cycles at the last follow-up date, and the disease remained stable. Next-generation sequencing (NGS) has revolutionized genomic research on many hematologic cancer types. However, genomic data on primary cutaneous lymphomas are still incomplete and sometimes contradictory [33]. Therefore, clinicians still rely on clinical and histologic information rather than genetic diagnostics for diagnosis and determining the choice of therapy. The identification of genes and pathways involved in the pathogenesis of different pcPTCL-NOS subtypes can lead to novel targeted therapies in oncology [34].

## 5. Conclusions

In conclusion, patients with pcPTCL-NOS present tumors with rapid growth, and the disease prognosis is poor. We reported a female predominance, and the median age at diagnosis was 54 years. The most common clinical presentations were nodules/tumors and papules, and, less often, ulcers; lesions in the head were common. pcPTCL-NOS may require early and active systemic treatment. However, reducing the intensity of treatment with CHOP should be appropriately considered for patients with T1 disease to avoid overtreatment. Our study has some limitations. (1) This study has a small sample size, which limits its statistical power. (2) This study has a retrospective design. (3) This study has a referral bias to a tertiary center for patients with advanced-stage disease and treatment heterogeneity. pcPTCL-NOS is very rare; therefore, more studies on pcPTCL-NOS are needed to better characterize the tumors and clarify the best treatment modality.

## Figures and Tables

**Figure 1 cancers-17-01673-f001:**
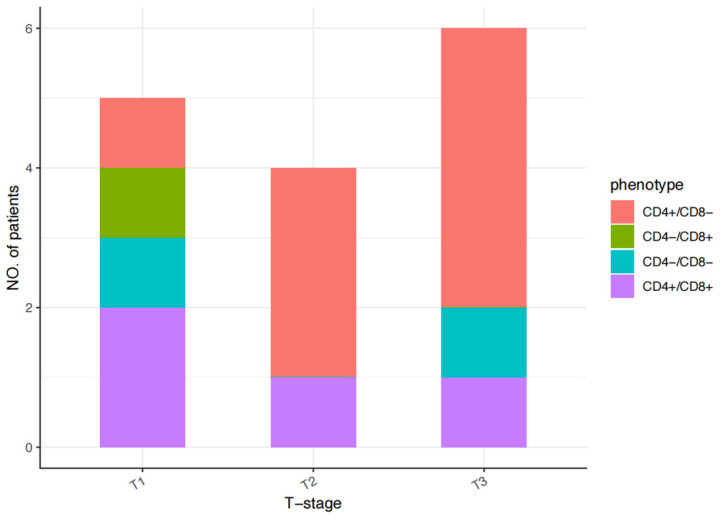
CD4/CD8 immunophenotypic characteristics of each patient with pcPTCL-NOS.

**Figure 2 cancers-17-01673-f002:**
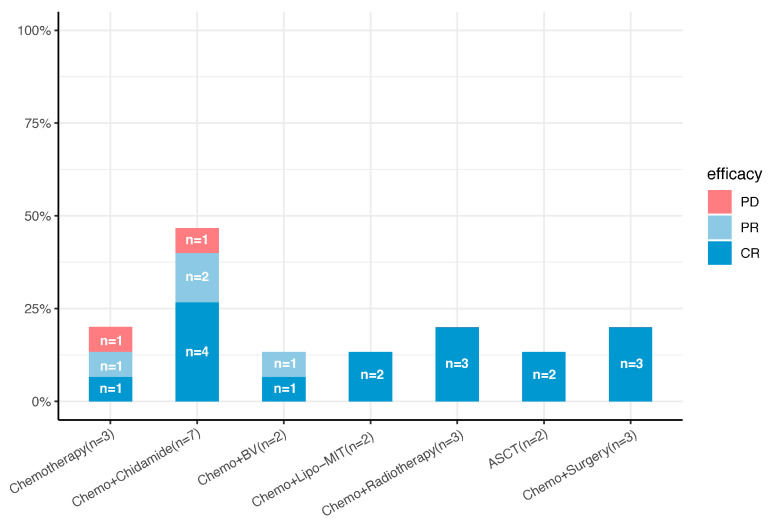
Front-line treatment plans and efficacy for 15 patients with pcPTCL-NOS (n, number; Chemo, chemotherapy; Lipo-MIT, mitoxantrone liposomes; BV, brentuximab vedotin; ASCT, autologous stem cell transplantation; CR, complete response; PR, partial response; PD, progressive disease).

**Figure 3 cancers-17-01673-f003:**
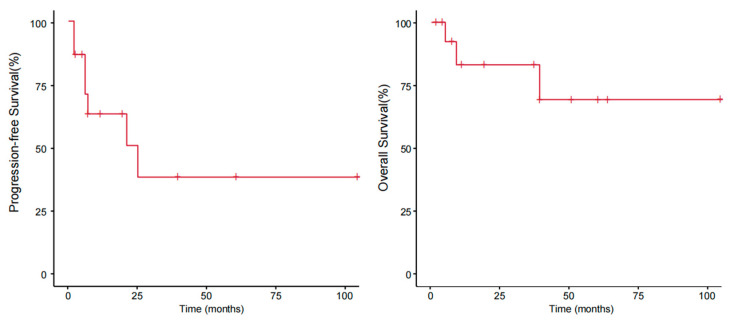
Progression-free survival and overall survival of patients with pcPTCL-NOS.

**Figure 4 cancers-17-01673-f004:**
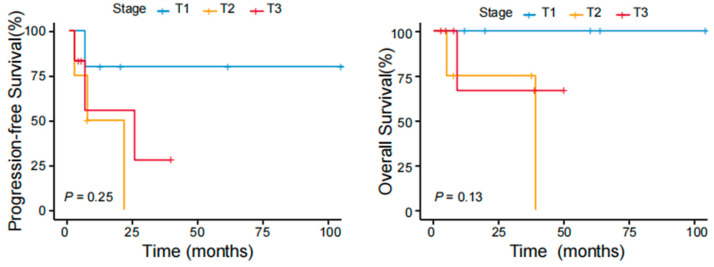
PFS and OS of patients with pcPTCL-NOS stratified according to different tumor stages.

**Figure 5 cancers-17-01673-f005:**
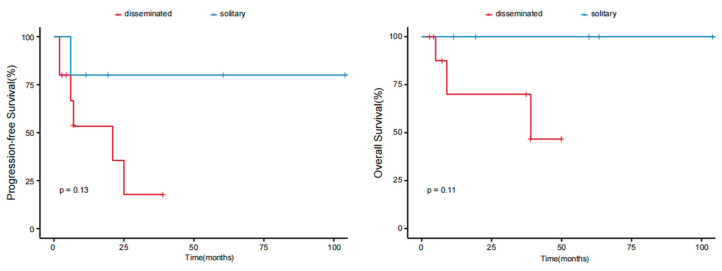
PFS and OS of patients with pcPTCL-NOS stratified according to different body regions.

**Table 1 cancers-17-01673-t001:** Clinical features of each patient with pcPTCL-NOS.

Case Number	Age	Sex	T Stage	Clinical Presentation	B Symptoms	Serum β2-MG Level	Serum LDH Level
1	54	F	T1	Nodule	Yes	Normal	Normal
2	38	F	T1	Nodule	No	Normal	Normal
3	28	F	T1	Nodule	No	Normal	Normal
4	54	F	T1	Nodule	No	Normal	Normal
5	66	F	T1	Nodule	No	Normal	Normal
6	61	F	T2	Nodule	No	Elevated	Normal
7	52	F	T2	Nodule and ulcers	No	Normal	Elevated
8	66	F	T2	Papules	No	Normal	Elevated
9	72	F	T2	Nodule	Yes	Normal	Elevated
10	31	M	T3	Nodule	No	Elevated	Normal
11	32	M	T3	Nodule and ulcers	No	Normal	Normal
12	66	M	T3	Papules	No	Elevated	Normal
13	54	M	T3	Papules	No	Elevated	Elevated
14	32	F	T3	Papules	No	Normal	Normal
15	58	F	T3	Nodule	No	Normal	Normal

The table summarizes the data according to the tumor stage at diagnosis: T1 stage (case nos. 1–5; columns highlighted in yellow), T2 stage (case nos. 6–9; columns highlighted in blue), and T3 stage (case nos. 10–15; columns highlighted in gray). Abbreviations: β2-MG, β2-microglobulin; LDH, lactate dehydrogenase.

**Table 2 cancers-17-01673-t002:** Histological and immunophenotypic features of each patient with pcPTCL-NOS.

Case Number	Ki67 (%)	CD20	CD30	PD-1	CD4+/CD8−	CD4−/CD8+	CD4−/CD8−	CD4+/CD8+	EBER	EBV DNA
1	70%	Negative	Negative	Negative			+		Negative	Negative
2	30%	Positive	Negative	Negative				+	Negative	Negative
3	90%	Negative	Negative	Positive		+			Negative	Negative
4	>80%	Negative	Positive	Negative	+				Negative	Negative
5	>40%	Positive	Positive	Negative				+	Negative	Negative
6	90%	Negative	Negative	Negative	+				Negative	Negative
7	60%	Negative	Positive	Negative	+				Negative	Negative
8	30%	Positive	Positive	Negative				+	Negative	Negative
9	80%	Negative	Negative	Negative	+				Negative	Negative
10	30%	Negative	Negative	Positive				+	Negative	Negative
11	70%	Negative	Positive	Negative	+				Negative	Negative
12	50%	Positive	Negative	Negative			+		Negative	Positive
13	50%	Negative	Negative	Negative.	+				Negative	Negative
14	80%	Positive	Negative	Negative	+				Negative	Negative
15	80%	Negative	Positive	Positive	+				Negative	Negative

The table summarizes the data according to the tumor stage at diagnosis: T1 stage (case nos. 1–5; columns highlighted in yellow), T2 stage (case nos. 6–9; columns highlighted in blue), and T3 stage (case nos. 10–15; columns highlighted in gray). Abbreviations: EBV, Epstein–Barr virus. EBER, Epstein–Barr virus-encoded small RNA.

**Table 3 cancers-17-01673-t003:** First-line and salvage treatments, responses, and survival of each patient with pcPTCL-NOS.

Case Number	T-Stage	First-Line Therapy				Radiotherapy	ASCT	Surgery	Initial Response to Treatment	Salvage Therapy	Final Response to Treatment	Survival	PFS Time (Months)	OS Time (Months)
		Chemotherapy	Chidamide	Brentuximab Vedotin	Lipo-MIT									
1	T1	CHOPE × 6	No	No	No	Yes	No	Yes	CR	Chidamide + Azacitidine × 6	CR	alive without disease	6	63
2	T1	CMOPE × 6	No	No	Yes	No	No	No	CR		CR	alive without disease	11	11
3	T1	CHOP × 4	Yes	No	No	No	No	Yes	CR		CR	alive without disease	19	19
4	T1	CHOPE × 6	Yes	No	No	Yes	No	No	CR		CR	alive without disease	60	60
5	T1	CHOP × 6	No	No	No	No	No	Yes	CR		CR	alive without disease	105	105
6	T2	CHOPE × 6	Yes	No	No	No	No	No	PD	GDP × 1, Chidamide + DICE × 4	CR	alive without disease	7	37
7	T2	CHOPE × 6	No	No	No	No	No	No	PR	GDP × 4	PD	death due to lymphoma	21	39
8	T2	CHOP × 1 + CHP × 3	No	No	No	No	No	No	PR	Chidamide + BV + CHP × 4	PR	alive with disease	7	7
9	T2	CHOPE × 3	Yes	No	No	No	No	No	PR	HD-MTX × 1, DHAP × 1	PD	death due to lymphoma	2	5
10	T3	CHOPE × 6	Yes	No	No	Yes	Yes	No	CR	Tislelizumab × 18	CR	alive without disease	25	50
11	T3	CHOPE × 4 + CHP × 2	No	Yes	No	No	Yes	No	CR		CR	alive without disease	39	39
12	T3	CHOPE × 2	No	No	No	No	No	No	PD	Chidamide + DICE × 1, Chidamide + Azacitidine × 5, GemOx × 1	PD	death due to lymphoma	2	9
13	T3	CHOP × 6	Yes	No	No	No	No	No	CR	Chidamide + Golidocitnib	SD	alive with disease	6	7
14	T3	CMOPE × 6	No	No	Yes	No	No	No	CR		CR	alive without disease	4	4
15	T3	CHP × 4	Yes	Yes	No	No	No	No	PR		PR	alive with disease	3	3

Abbreviations: CHOP (cyclophosphamide, doxorubicin, vincristine, and prednisone); CHOPE (cyclophosphamide, doxorubicin, vincristine, prednisone, and etoposide); CMOPE (cyclophosphamide, mitoxantrone liposome, vincristine, prednisone, and etoposide); GDP (gemcitabine, dexamethasone, and cisplatin); DICE (dexamethasone, ifosfamide, cisplatin, and etoposide); BV (brentuximab vedotin); CHP (cyclophosphamide, doxorubicin, and prednisone); HD-MTX (high-dose methotrexate); GemOx (gemcitabine and oxaliplatin); DHAP (dexamethasone, cytarabine, cisplatin; ASCT (autologous stem cell transplantation)).

**Table 4 cancers-17-01673-t004:** Univariate logistic regression analysis of the predictive factors for progression-free survival.

Characteristics	Hazard Ratio (95% CI)	*p* Value
Age		0.395
≤60 years	1.0 (reference)	
>60 years	1.92 (0.43–8.66)	
Sex		0.488
Female	1.0 (reference)	
Male	0.58 (0.13–2.69)	
Stage		
T1	1.0 (reference)	
T2	2.73 (0.55–13.63)	0.222
T3	1.42 (0.31–6.57)	0.651
B symptoms		0.024
No	1.0 (reference)	
Yes	10.04 (1.36–7–4)	
PIT score		
0	1.0 (reference)	
1	2.66 (0.59–11.97)	0.203
2	2.06 (0.23–18.48)	0.519
3	NA	NA
4	NA	NA
B2-microglobulin level		0.063
Normal	1.0 (reference)	
Elevated	4.14 (0.92–18.57)	
Ki67		0.902
<80%	1.0 (reference)	
≥80%	1.11 (0.17–4.73)	
CD20 expression		0.411
Negative	1.0 (reference)	
Positive	0.41 (0.05–3.43)	
CD30 expression		0.101
Negative	1.0 (reference)	
Positive	0.17 (0.02–1.41)	
PD-1 expression		0.403
Negative	1.0 (reference)	
Positive	2.25 (0.34–14.97)	
Immunophenotype		
CD4+/CD8−	1.32 (0.29–5.96)	0.715
CD4−/CD8+	0 (0–Inf)	0.999
CD4−/CD8−	10.04 (1.36–74)	0.024
CD4+/CD8+	0.27 (0.03–2.23)	0.222
Treatment regimen		
Chidamide	1.6 (0.36–7.18)	0.537
Brentuximab vedotin	0 (0–Inf)	0.999
Lipo-MIT	0 (0–Inf)	0.999
Radiotherapy	0.96 (0.18–5.16)	0.96
ASCT	0.46 (0.05–4.1)	0.487
Surgery	0.56 (0.07–4.64)	0.589

Abbreviations: PIT, prognostic index for T-cell lymphoma; Lipo-MIT, mitoxantrone liposome; ASCT, autologous stem cell transplantation; NA, Not Applicable.

## Data Availability

The original contributions presented in this study are included in the article/Appendix A. Further inquiries can be directed to the corresponding authors.

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
