# Peer review of "Clinical Features and Outcomes of Primary Cutaneous Peripheral T-Cell Lymphoma, Not Otherwise Specified, Treated with CHOP-Based Regimens"

_cancers, 2025, doi:10.3390/cancers17101673_

Round 1
Reviewer 1 Report
Comments and Suggestions for Authors
The study presents valuable real-world data on an exceptionally rare and aggressive malignancy—primary cutaneous peripheral T-cell lymphoma, not otherwise specified (pcPTCL-NOS). It is based on a 10-year single-institution cohort, incorporating detailed clinical and pathological correlations. The comprehensive analysis of treatment responses and survival outcomes offers meaningful insights for clinical practice, particularly in the absence of prospective trials.
While the manuscript is generally well-structured, several areas require improvement to enhance its clarity, analytical depth, and scientific rigor:
- Keywords should be revised to avoid redundancy with terms already present in the title.
- Despite acknowledgment of the disease’s rarity, the limited sample size (n=15) substantially restricts the generalizability of the findings.
- The analysis includes only univariate Cox regression; the addition of a multivariate model is strongly recommended to identify independent predictors of progression-free and overall survival.
- Although immunophenotypic characterization is emphasized, the absence of molecular profiling (e.g., NGS or cytogenetics) limits the study’s translational relevance. The inclusion of such data would significantly enhance disease classification, prognostication, and therapeutic guidance.
- The figures are of low resolution and lack adequate labeling. Improved visualization and the inclusion of informative captions with relevant clinical implications are advised.
While generally readable, the manuscript would benefit from professional English editing to polish grammar, sentence structure, and flow.
Author Response
Comments 1. Keywords should be revised to avoid redundancy with terms already present in the title.
Reply: Thank you very much for your valuable comments. According to your advice, we have revised keywords in the text.
Changes in the text:
Comments 2. Despite acknowledgment of the disease’s rarity, the limited sample size (n=15) substantially restricts the generalizability of the findings.
Reply: We agree with this comment. We have further fully explained the limitations of our work within the text to avoid exaggerating the research results.
Changes in the text:
We added ‘Our study has some limitations. (1) This study has a small sample size, which limits its statistical power. (2) This study has a retrospective design. (3) This study has a referral bias to a tertiary center for patients with advanced-stage disease and treatment heterogeneity.’in the Conclusions section (page 14, lines 416–419 in our revised manuscript).
Comments 3. The analysis includes only univariate Cox regression; the addition of a multivariate model is strongly recommended to identify independent predictors of progression-free and overall survival.
Reply: Thank you for your guidance.We agree with this comment. Due to the limited sample size, there was considerable bias in the conduct of multivariate analysis. However, for the rarity of pcPTCL-NOS, this is also an exploratory research.
Comments 4. Although immunophenotypic characterization is emphasized, the absence of molecular profiling (e.g., NGS or cytogenetics) limits the study’s translational relevance. The inclusion of such data would significantly enhance disease classification, prognostication, and therapeutic guidance.
Reply: Thank you for noting this issue, and we agree with this comment. At present, our study lacks a molecular profiling analysis, which is also the goal and direction of our next work. We have added relevant contents in the Discussion section.
Changes in the text:
Comments 5. The figures are of low resolution and lack adequate labeling. Improved visualization and the inclusion of informative captions with relevant clinical implications are advised.
Reply: Thank you for noting this issue. In accordance with your advice, we have revised the figures and added informative captions.
Changes in the text:
Comments 6. the Quality of English Language
While generally readable, the manuscript would benefit from professional English editing to polish grammar, sentence structure, and flow.
Reply: Thank you for noting this issue. We have improved the manuscript through a professional English editing service(American Journal Experts), which polished the grammar, sentence structure, and flow. The manuscript is now better presented.

Reviewer 2 Report
Comments and Suggestions for Authors
Review of: Clinical Features and Outcomes of Primary Cutaneous Peripheral T-Cell Lymphoma, Not Otherwise Specified, Treated with CHOP-Based Regimens
GENERAL COMMENTS:
- THE DATA / RESULTS ARE PRESENTED MULTIPLE TIMES IN THE TEXT AND REPEATED IN THE DISCUSSION. IT IS HARD TO MAKE SENSE OF THE IMPORTANT PEARLS OF INFORMATION DUE TO THE PRESENTATION. THE TABLES AND KM CURVES HELP, BUT TO ME THIS COMES ACROSS AS A COMBINATION OF MULTIPLE CASE REPORTS. CASE REPORTS ARE VALUABLE AND AS SUCH THE DATA IS VALUABLE BUT THE PRESENTATION NEEDS TO BE CONDENSED AND IMPROVED.
- THE ANALYSIS FOR SIGNIFICANCE IN OUTCOME MAY NOT BE VALID DUE TO THE EFFECT OF MULTIPLE STATISTICAL ANALYSES ON A SMALL DATA SET. SEE ARTICLE BELOW https://pmc.ncbi.nlm.nih.gov/articles/PMC4840791/
SPECIFIC COMMENTS:
LINE 29-32, TABLE S1, AND THROUGHOUT THE TEXT:
“CD4+/CD8- 53.33%; 29 CD4+/CD8+ 26.67%; CD4-/CD8- 13.33%; and CD4-/CD8+ 6.67%. One patient had a T follicular helper (TFH) phenotype, with CD10+, CXCL13+, PD1+, and BCL6+ phenotypes. 31 Moreover, 33.33%”
THE USE OF MORE THAN TWO OR PERHAPS THREE SIGNIFICANT FIGURES DOES NOT ADD VALUE.
LINE 34 “patients undergoing treatment combined with chidamide,”
PLEASE INCLUDE GENERIC NAME AT FIRST USE (Tucidinostat)
LINE 120 “had B symptoms, and elevated beta 2-microglobulin (β2-MG) and lactate dehydrogenase (LDH)
ADD ABBRIEVIATION AT FIRST USE
LINE 144 – 145: “The presence of EBV as assessed 144 by in situ hybridization for EBER was negative in all patients, with one patient in stage T3 145 testing positive for EBV-DNA
PLEASE DEFINE EBER ABBREVIATION
TABLE 2 ON PAGE 4: THE POSITIVES ARE HIDDEN BY THE NEGATIVES. PERHAPS HILIGHTING THE POSITIVES WOULD MAKE THEM MORE VISISBLE
LINE 168-9: Notably, two patients received the CMOPE 168 regimen, in which mitoxantrone liposomes (Lipo-Mit) replaced anthracyclines in the 169 CHOPE regimen,
WHAT IS THE E IN THESE REGIMENS? ETOPOSIDE??
LINE 191 “second-line chemotherapy, such as the GDP, GEMOX, 191 DICE, and DHAP”
OK I HAVE DECIDED YOU NEED TO PUT IN A SUPPLIMENTARY TABLE OF ALL THE REGIMENS AND THE AGENTS IN THAT REGIMEN AND MIGHT AS WELL INCLUDE CHOP
|
REGIMEN |
AGENTS |
|
CHOP |
Cyclophosphamide, Doxorubicin, Vincristine, Prednisone |
LINE 219: Four patients with short follow-up times were excluded, and a total of 11 patients were included in the survival analysis.
YOU CANNOT EXCLUDE PATIENTS FROM SURVIVAL CURVE
TABLE 4: THE P-VALUE COLUMN DOES NOT SEEM TO LINE UP CORRECTLY WITH OTHER COLUMNS.
PLEASE DEFINE PIT SCORE ELEMENTS.
FIGURE 4 AND FIGURE 5: THE TICK MARKS ON THE CURVES ARE TOO SMALL TO BE SEEN
Author Response
Responses to the comments from the reviewers
Reviewer #2
Comments 1. The data / results are presented multiple times in the text and repeated in the discussion. It is hard to make sense of the important pearls of information due to the presentation. The tables and KM curves help, but to me this comes across as a combination of multiple case reports. Case reports are valuable and as such the data is valuable but the presentation needs to be condensed and improved.
The analysis for significance in outcome may not be valid due to the effect of multiple statistical analyses on a small data set. See article below https://pmc.ncbi.nlm.nih.gov/articles/PMC4840791/
Reply: Thank you very much for your valuable comments. According to your advice, we have removed duplicates from the text. We have carefully studied the references you suggested and have made corresponding modifications to the data analysis section.
Changes in the text: The changes were made as follows:
Comments 2. Line 29-32, table S1, and throughout the text: “CD4+/CD8- 53.33%; 29 CD4+/CD8+ 26.67%; CD4-/CD8- 13.33%; and CD4-/CD8+ 6.67%. One patient had a T follicular helper (TFH) phenotype, with CD10+, CXCL13+, PD1+, and BCL6+ phenotypes. 31 Moreover, 33.33%” The use of more than two or perhaps three significant figures does not add value.
Reply: Thank you for your meticulous review and for bringing this oversight to our attention. The Abstract was modified to describe this information concisely and clearly.
Changes in the text:
Comments 3. Line 34 “patients undergoing treatment combined with chidamide,” Please include generic name at first use (tucidinostat)
Reply: Thank you for noting this issue. We agree with this comment. We have added the generic name at the first use (tucidinostat).
Changes in the text:
The term “(tucidinostat)” was added to the Abstract (page 2, line 46 in our revised manuscript).
Comments 4. LINE 120 “had B symptoms, and elevated beta 2-microglobulin (β2-MG) and lactate dehydrogenase (LDH) . Add abbrieviation at first use
Reply: Thank you for noting this issue. We agree with this comment. We have added abbreviations at their first use.
Changes in the text:
Comments 5. LINE 144-145:“The presence of EBV as assessed by in situ hybridization for EBER was negative in all patients, with one patient in stage T3 testing positive for EBV-DNA. Please define EBER abbreviation
Reply: Thank you for noting this issue. We agree with this comment and have added the abbreviations.
Changes in the text:
Comments 6. TABLE 2 ON PAGE 4: The positives are hidden by the negatives. Perhaps hilighting the positives would make them more visisble.
Reply: Thank you for noting this issue. We agree with this comment. We used different colors for patients with different disease stages.
Changes in the text:
Comments 7. LINE 168-9: Notably, two patients received the CMOPE regimen, in which mitoxantrone liposomes (Lipo-Mit) replaced anthracyclines in the CHOPE regimen,What is the E in these regimens? Etoposide??
Reply: Thank you for noting this issue. This error is due to our negligence. E is the abbreviation for etoposide.
Changes in the text:
Comments 8. LINE 191 “second-line chemotherapy, such as the GDP, GEMOX, DICE, and DHAP”
Ok I have decided you need to put in a supplimentary table of all the regimens and the agents in that regimen and might as well include CHOP
|
REGIMEN |
AGENTS |
|
CHOP |
Cyclophosphamide, Doxorubicin, Vincristine, Prednisone |
Reply: Thank you for noting this issue. We have included all the regimens in the table as follows:
|
REGIMEN |
AGENTS |
|
CHOP |
cyclophosphamide, doxorubicin, vincristine, and prednisone |
|
CHOPE |
cyclophosphamide, doxorubicin, vincristine, prednisone and etoposide |
|
CMOPE |
cyclophosphamide, mitoxantrone liposome, vincristine, prednisone and etoposide |
|
GDP |
gemcitabine, dexamethasone, and cisplatin |
|
DICE |
dexamethasone, ifosfamide, cisplatin, and etoposide |
|
CHP |
cyclophosphamide, doxorubicin, and prednisone |
|
Gemox |
Gemcitabine and oxaliplatin |
|
DHAP |
dexamethasone, cytarabine, and cisplatin; ASCT, autologous stem cell transplantation |
Changes in the text:
Comments 9. LINE 219: Four patients with short follow-up times were excluded, and a total of 11 patients were included in the survival analysis.
You cannot exclude patients from survival curve.
Reply: Thank you for noting this issue. As suggested, we redrew the survival curve for all 15 patients.
Changes in the text:
Comments 10. TABLE 4: The p-value column does not seem to line up correctly with other columns. Please define PIT score elements.
Reply: Thank you very much for your valuable comments. In accordance with your advice, we adjusted Table 4 to ensure that the p value column lines up correctly with the others and added a more detailed interpretation of the PIT score.
Changes in the text:
- The following text was added to the Results section: ‘(calculated from 0 to 4 by age, performance status, lactic dehydrogenase level, and bone marrow involvement)’(page 10, lines 267–268 in our revised manuscript).
Comments 11. Figure 4 and Figure 5: The tick marks on the curves are too small to be see.
Reply: Thank you for noting this issue. We updated Figures 4 and 5 in the text.
Changes in the text:

Reviewer 3 Report
Comments and Suggestions for Authors
This retrospective study investigates the clinical features, immunophenotypic profiles, treatments, and outcomes of 15 patients with primary cutaneous peripheral T-cell lymphoma, not otherwise specified (pcPTCL-NOS), treated at a single center between 2014 and 2024. All patients received CHOP-based chemotherapy regimens, with some receiving additional novel agents or radiotherapy. The study highlights the heterogeneity in clinical presentation and immunophenotype, noting poor outcomes particularly in patients with B symptoms, elevated β2-microglobulin, or a CD4-/CD8- phenotype. The findings suggest that while aggressive systemic treatment is typically required, patients with solitary (T1 stage) lesions might benefit from de-escalated therapy. This work contributes real-world data on an exceptionally rare lymphoma subtype.
Some suggestions:
-
Clearly highlight how this cohort adds to or differs from previous studies, particularly regarding treatment recommendations for stage T1 patients.
- Describe the inclusion and exclusion criteria more explicitly, especially how systemic PTCL-NOS was ruled out.
- Clarify the rationale for choosing a statistical significance threshold of p < 0.1 instead of the conventional p < 0.05.
- Integrate a more critical comparison with prior large cohort studies.
- Expand the limitations section to discuss referral bias, retrospective design, sample size, and possible treatment heterogeneity.
Author Response
Comments 1. Clearly highlight how this cohort adds to or differs from previous studies, particularly regarding treatment recommendations for stage T1 patients.
Reply: Thank you for your constructive suggestion. Compared with previous studies, we provided a detailed description of the selection of treatments for our cohort, and all patients were treated with CHOP-based regimens as the initial treatment, regardless of the tumor stage. As you suggested, we added relevant descriptions to the Discussion section.
Changes in the text:
Comments 2. Describe the inclusion and exclusion criteria more explicitly, especially how systemic PTCL-NOS was ruled out.
Reply: Thank you for your suggestion. Primary cutaneous lymphomas are defined as non-Hodgkin lymphomas that present in the skin with no evidence of extracutaneous disease at the time of diagnosis. As you suggested, we have added the corresponding description to the text.
Changes in the text:
Comments 3. Clarify the rationale for choosing a statistical significance threshold of p < 0.1 instead of the conventional p < 0.05.
Reply: We sincerely thank the reviewer for carefully reading our manuscript. We changed the P value to “< 0.05”, and the corresponding description in the text has been modified to avoid exaggerating the results.
Changes in the text:
Comments 4. Integrate a more critical comparison with prior large cohort studies.
Reply: Thank you for your suggestion. Owing to the rarity of pcPTCL-NOS, few case series exist, and most include patients with other histologies or patients with concomitant systemic disease. As suggested by the reviewer, we further increased the comparison with previous studies.
Changes in the text:
- The following text was added to the Discussionsection: ‘Two patients received CMOPE regimen as initial treatment,and achieved CR.This was a phase I study initiated by our center,and showed promising anti-tumor activity,evidenced by ORR of 100% and CR rate of 66.7% as first-line therapy in patients with untreated PTCL[31].’ (page 13, lines 363–366 in our revised manuscript).
- The following text was added to the Discussionsection: ‘PD-1, a membrane receptor expressed on activated T cells, inhibits the immune response in peripheral tissues and promotes self-tolerance. In CTCL, skin lesion-derived tumor-infiltrating T cells have higher expression of PD-1 (and other immune checkpoint molecules) in comparison to controls[34]. Nevertheless, ongoing studies should define more clearly the efficacy and safety of PD1 inhibitors in the T-cell lymphoma treatment.’ (page 14, lines 388–393 in our revised manuscript).
Reply: Thank you for your constructive suggestion. As you suggested, we have expanded the limitations section to include the referral bias, retrospective design, sample size, and possible treatment heterogeneity.
Changes in the text:
We added ‘Our study has some limitations. (1) This study has a small sample size, which limits its statistical power. (2) This study has a retrospective design. (3) This study has a referral bias to a tertiary center for patients with advanced-stage disease and treatment heterogeneity.’in the Conclusions section (page 14, lines 416–419 in our revised manuscript).

Round 2
Reviewer 1 Report
Comments and Suggestions for Authors
Recommended for publication in the present form.
Reviewer 2 Report
Comments and Suggestions for Authors
Thank you for your revisions which have greatly improved the manuscript